

# Nowcasting commodity prices using social media

Jaewoo Kim[1], Meeyoung Cha[1] and Jong Gun Lee[2]

[1] Graduate School of Culture Technology, Korea Advanced Institute of Science & Technology, Daejeon, South Korea
[2] United Nations Global Pulse

## ABSTRACT

Gathering up-to-date information on food prices is critical in developing regions, as it allows policymakers and development practitioners to rely on accurate data on food security. This study explores the feasibility of utilizing social media as a new data source for predicting food security landscape in developing countries. Through a case study of Indonesia, we developed a nowcast model that monitors mentions of food prices on Twitter and forecasts daily price fluctuations of four major food commodities: beef, chicken, onion, and chilli. A longitudinal test over 15 months of data demonstrates that not only that the proposed model accurately predicts food prices, but it is also resilient to data scarcity. The high accuracy of the nowcast model is attributed to the observed trend that the volume of tweets mentioning food prices tends to increase on days when food prices change sharply. We discuss factors that affect the veracity of price quotations such as social network-wide sensitivity and user influence.

## INTRODUCTION

The ability to rapidly monitor food price fluctuations is critical to government institutions, production companies, and investment banks for making agile policy decisions and managing risks (*Cavallo, 2013*; *Shaun & Lauren, 2014*). The demand for data has increased in a hyperconnected world, where countries, markets, and people affect one other in a complex manner (*Pentland, 2014*). However, not all countries have the capability to monitor high-resolution commodity price data. Some developing countries publish official commodity price data at a slower rate, sometimes monthly or quarterly. This significant delay in releasing economic indicators is largely due to the lack of infrastructure to gather market data (*Aizenman & Marion, 1999*). In fact, political and financial reasons have hindered a few countries from publishing the consumer price indexes for several decades (*Grosh & Glewwe, 2000*). Nonetheless, because commodity price and in particular food insecurity in developing regions is extremely dynamic[1], the ability to track market status quickly and to predict food commodity price trends is an all the more critical challenge (*Gouel, 2013*).

---

[1] Food insecurity in developing regions are a severe problem and the rice price in Haiti surged by 81% in 2008 alone.

Corresponding authors
Meeyoung Cha,
meeyoungcha@kaist.ac.kr
Jong Gun Lee, jonggun.lee@un.or.id

Remarkable progress has been made over the last decade in acquiring market data. First via access to new technology. According to the International Telecommunication Union, there are more than seven billion mobile cellular subscriptions in the world, corresponding to a global penetration rate of 97%. Such technology enables developing countries to attain a level of financial data access that until recently was only possible in more developed economies. Second is the innovative proposals and methods that fill information gaps and track economic data better in places where standard approaches cannot be easily applied. For instance, price indexes constructed from the Web (such as online shopping sites that directly cite commodity prices) can produce alternative inflation estimates (*Cavallo, 2013*). Crowdsourcing is another such approach, where price quotations reported by individuals are collected and analyzed in initiatives like (*Premise, 2016*). In Nigeria and India, microeconomic databases of consumer goods were successively built by combining scrapers for online e-commerce data with a crowd-sourced data via mobile applications (*Liz, 2013*). Price collectors in this system comprise retailers and non-professional volunteers, who receive compensation in various forms of rewards like money and communication credit. The World Bank has also conducted a pilot study for crowd-sourced price data collection through mobile phones and non-professional price collectors (*Hamadeh, Rissanen & Yamanaka, 2013*). Price data was collected for thirty tightly specified food commodity items on a monthly basis for approximately six months in eight pilot countries.

Recently an alternative source of information has become widely available as a new economic signal (*Pappalardo et al., 2016*). User-generated data from various online social network services (OSNs) have been a source of indicative signals for predicting various societal phenomena including human behavior in crisis situations (*Vieweg, Olteanu & Castillo, 2015*), economic market changes (*Bollen, Mao & Zeng, 2011*; *Asur et al., 2010*), and flu trends (*Lampos et al., 2015*; *Ginsberg et al., 2009*). Utilizing large-scale OSN signals has several benefits. First, social network signals are less costly than crowdsourcing because there is no need to reward individuals who generate data (*Simula, 2013*). Second, the continuous nature of OSN data allows for near real-time monitoring or what is called *nowcasting* (*Giannone, Reichlin & Small, 2008*).

Designing a nowcast model for commodity prices, however, is a complex problem. This is because the task needs to produce accurate estimates of the official commodity prices, provide early warning signals of unexpected spikes in the real world, and adapt to a variety of commodities for wider applicability (*Lampos & Cristianini, 2012*). These goals are harder to achieve in developing countries, where economic status is volatile and social media is less widely used. Nonetheless, rapidly expanding Web infrastructure, supported by humanitarian projects that provide free Internet in rural areas such as Internet.org (*Facebook, 2016*), is being observed in many developing countries (*Ali, 2011*) and social media data can hence serve as an additional, non-invasive measurement method for those regions.

This paper presents a case study of adopting micro-blogging platform signals on Twitter as an additional data source for building a food price nowcast model in Indonesia. This research was initiated by the government of Indonesia as part of its effort to combine and adopt different sources of information to produce highly credible market statistics. Four

critical food commodities (beef, chicken, onion, and chilli) were chosen as the first set of items to be tracked based on national food security priorities and data availability. Twitter was chosen as a data source because of its popularity within the country; Indonesia has one of the highest adoption rates in the world for Twitter, both in terms of number of users and amount of generated content.

The main goal of this work is to create a nowcast model that reproduces time series of daily prices for the four chosen commodities during a 15-month investigation period between June 2012 and September 2013 based solely on price information from tweets. This main goal is achieved by three specific aims. First, the model should be able to provide price time series that highly correlate with real-world price trends. We conduct an evaluation by using the Pearson correlation coefficient to determine a correlation between an official and a predicted price time series. Secondly, the model should be able to estimate the absolute price value with minimized error in daily scale. We conduct the evaluation by using mean absolute percentage error (MAPE) to evaluate a magnitude of error between an official and a predicted price time series. Thirdly, the model should be capable of nowcasting food price, which is defined as capturing information on a real-time basis within a short time gap typically in the single day range. For checking the feasibility of using the model as a daily price predictor, we conduct an additional evaluation process by using a cross-correlation coefficient (CCF) that could estimate how an official and a predicted time series are related at different time lags. We have shown that those predicted time series have the highest correlation at a lag within the timeframe of a single day, therefore we could clarifies that the price time series produced by the model is able to be used for nowcasting.

A two-step algorithm is proposed in this research. In the first step, a keyword filter is used to extract tweets mentioning price quotations of the four food commodities from the entire corpus of tweets that were generated from Indonesia between June 2012 and September 2013, a timeframe of 15 months. A numerical model parameter is also used to filter the tweets to ensure that the tweet price does not exceed a maximum allowable daily percentage price change (computed based on historical rates). The keyword and numerical filters extracted 41,761 relevant tweets from the data. In the second step, a statistical model, using OSN data, is built to accurately estimate food prices for each commodity in order to assist with the official statistics publicized by the Indonesian government. The nowcast model produces estimates of commodity prices that have a high correlation with official food price statistics over the timeframe covered and shows better prediction performance than existing algorithms. This paper also describes the effect of several important social network-wide variables, via testing the robustness of the model under data scarcity conditions and by modeling user-level credibility to suggest an enhanced sampling strategy.

This research finds that Indonesians do tweet about food prices, and that those prices closely approximate official figures. A near real-time food price index that is nowcasted using social media signals may be an efficient tool with immediate utility for policy makers and economic risk managers. The results of this study are being used as a basis for the development of OSN-assisted nowcast systems in several other developing countries under

**Table 1  Full keyword taxonomy for tweet collection.**

| | | |
|---|---|---|
| Commodity names | Beef | ("sapi") |
| | Chicken | ("daging") AND ("ayam") |
| | Onion | ("bawang") |
| | Chilli | ("cabe"\| "cabai") |
| Prices | Values | (Digits) AND ("rb" \| "ribu" \| "ratus" \| ",-" \| ",00" \| None) |
| | Units | ("rp" \| "rupiah") |
| Commodity units | | ("per" \| "se") AND (Letters) |

the United Nations World Food Programme (WFP). Details of this research, including the online demo, are available at http://www.unglobalpulse.org/nowcasting-food-prices.

# METHODS

## Data collection

Indonesia is a good testbed for this study for two reasons. First, reliable ground-truth data is available on a daily basis. The Ministry of Trade in Indonesia collects and publishes daily price information, which is also published as monthly records by the Bureau of Statistics. Second, social media, like Twitter, are widely used in the country so that there are enough online signals on commodity prices. In fact, Indonesia is one of the top-five tweeting countries (*Siim, 2013*).

Four basic food commodities, beef, chicken, onion, and chilli, were chosen for monitoring based on the availability of data in terms of tweet mentions and the country-level priorities for food security monitoring in consultantation with the Ministry of National Development Planning (Bappenas) and the WFP in Indonesia. Beef and chicken are in fact the two most commonly consumed meats in Indonesia, as people rarely consume pork. Likewise onion and chilli are the most popular spices across the nation. As a result, prices of these four commodity items have been frequently utilized to monitor inflation, where chilli in particular has been considered sensitive to inflation (*Amindoni, 2016*; *Sawitri, 2017*). Daily food price data can be obtained for these four target commodities via the webpage of the Ministry of Trade of Indonesia: https://ews.kemendag.go.id/.

Tweets were collected through a firehose access to Twitter, which returns a complete set of data. We screen for price mentions between June 2012 and September 2013, for 15 months. A taxonomy of keywords and phrases in Bahasa (i.e., the official language in Indonesia) is developed and used. The full taxonomy is mostly composed of commodity names, prices, and units (Table 1). Price information can be expressed in different ways, containing variations related to expressions of commodity name and mentioning prices. Price quotations are often mentioned in tweets with prefix Rp or suffix rupiah, where the price value may be either number or text. Commodity unit is also important; for instance expressions such as *per kilogram* or *per liter* are commonly used to define food price. Instead of using hundreds of regular expressions for normalizing various types of units into an identical unit, we suggest a nowcast model which can handle a commodity unit difference issue via a numerical approach. For the target commodities under this study,

**Table 2  Top-ten accounts with the largest tweet volume are all involved in advertising via bots.**

| Account name | Tweet volume | Attribute |
| --- | --- | --- |
| susu********** | 18,018 (22.95%) | Milk Ad |
| adhi******* | 216 (0.28%) | Distributor Ad |
| Ayam******* | 179 (0.23%) | Chicken Ad |
| kaos******* | 178 (0.23%) | Distributor Ad |
| Will********** | 169 (0.22%) | Milk Ad |
| bati********** | 166 (0.21%) | Distributor Ad |
| Grac******* | 162 (0.21%) | Dairy Ad |
| pull****** | 152 (0.19%) | Chicken Ad |
| keri******* | 123 (0.16%) | Farm Ad |
| indg********** | 108 (0.14%) | Meet Ad |

most price information from Twitter contains standardized units that are identical to the units of government official data, therefore it is possible to handle unit difference issue via numerical approach solely. Our model decides whether a commodity unit referenced in a tweet is appropriate or not by comparing its price value and credible price range.

Keyword combination for tweet collection:
(Commodity Names) AND (Price Values) AND (Price Units | Commodity Units)

As a result, a total of 78,518 tweets from 28,800 accounts were collected over the 15-month period. Below is an example tweet mentioning beef price and its translation in English:

Harga Daging Masih Rp 95 Ribu/Kg, Ini Cara Pemerintah Menekannya...
(Beef prices are still 95,000 Rupia per kilogram, this situation is pressing government...)

## Data cleaning

Tweet data contain noisy information and need to be cleaned prior to analysis. We employed the following measures in data cleaning. The first involves removing ambiguity in meaning. An obvious case of ambiguity arises when a single tweet quotes the price of two or more commodity items. Such cases occur in the 5% (2,607 times) of the entire price quotation data and were removed in advance of further investigation. Another case of ambiguity arises when the mentioned price is in relative terms, not in absolute terms (e.g., "price increased by X amount"). For instance, the word 'naik' in Indonesia means 'increase (up to)' or 'by'. Our data shows that price quotations containing the 'naik' word resulted in extremely small price ranges compared to the rest of the data. Hence, we removed tweet data containing this word, which accounted for 8% of the data.

Another important data cleaning task focuses on removing redundant messages or spam bots. Certain bot accounts can be identified based on their large quantity of duplicated tweets. We assume accounts that posted more than 100 tweets with over 80% of duplicated messages are bots. Table 2 shows the list of the-top ten bot accounts that mention prices the most frequently. Most accounts with large tweet volumes posted the price information of their products with the purpose of advertisement. This finding indicates that the majority

of accounts with a large volume of food price-related tweets are sellers. Note that the most prominent single account occupies 18,018 tweets (23% of all price quote tweets and 87% of all milk-related tweets). We can judge this account as a bot that promotes goat milk products, since its tweets are nearly identical to the following:

"sedia susu kambing etawa brand_name_hidden harga Rp 22 rb hub"
(Translation: Goat milk available for Rp 22000.)

We eliminate bot accounts from certain sellers which simply keep echoing the redundant content with a vast volume. In the following section, we suggest a model that utilizes the volume of a tweeted price to determine its credibility, and it seems not reasonable to assign more credibility to bot-tweeted information based on its proportion of volume than human-tweeted information. Previous studies have defined spam as a bot designed to give unfair influence on opinion by echoing the earlier information (*Chu et al., 2012*; *Lim et al., 2010*). The bots we define in this study act as a spam rather than play a valuable social role because they provide unfair and significant statistical bias to information distribution, therefore we employ a basic bot detection method to eliminate a high volume of redundant tweets.

As a result, we remove a total of 36,757 (46.8%) tweets from the data if (1) a tweet is an exact duplication of another (22.9%), (2) a tweet contains a specific word 'naik' describing the difference between two price values, like 'increased by' in English (6.5%), and (3) a tweet mentions more than one price (17.4%).

For the investigation period, the average number of tweets per account is 2.73. The contribution of tweets are heavily skewed among users so that the top-ten most prolific accounts posted 19,470 (24.8% of all) tweets. These top-ten accounts are all food vendors, e.g., local grocery shops advertising daily items (Table 2). In fact, people's motivations and willingness to post information on OSNs is influenced by external factors like news (*Gil de Zúñiga, Jung & Valenzuela, 2012*) or the interdependence of other industries (e.g., agriculture depends on machinery and transportation (*Richard, 2011*)). We find that people post more tweets during price-rising periods compared to price-decreasing periods. This tendency is more apparent with food commodities that have volatile price fluctuations and a smaller total volume of tweets—onion receives on average 2.8 times more tweets when prices are rising compared to price-decreasing periods.

## Price distribution

Once tweets mentioning prices are identified ($N = 41,761$), we may look into the price distributions. Figure 1A depicts example price quotations for onion on social media from a given day (translated in English) and the official price release of onion from the same date. Official price statistics are calculated from various vendor prices obtained from an off-line survey. Twitter signals have variations due to the geographic diversity of information sources, varying units, etc. These price quotations varied from one tweet to another and required data sensitization before they could be used for price prediction. Noise arises when commodity units are different (e.g., grams vs kilograms), mentions are of second-hand or related products (e.g., price of beef dishes instead of beef itself), or due to fake information, etc. Figure 1B shows the wide ranges of price quotations seen in raw social signals and

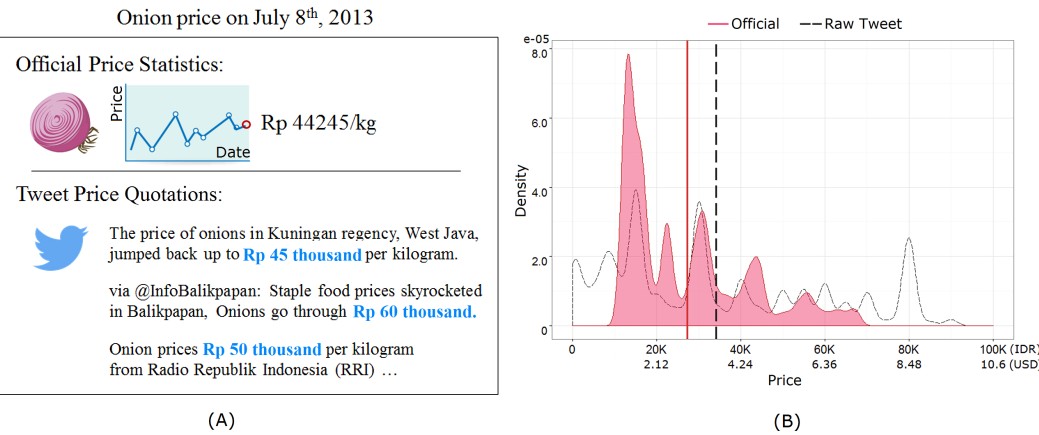

**Figure 1** (A) **The official price of onion on a given day and example price quotations from Twitter on the same day. (B) Official and tweet price distribution for onion over the monitored 15 months.** Distribution of raw price quotations from Twitter is denoted by the dashed line, while the solid red line is the official price published by the government for the same period. Vertical lines denote the mean values.

official prices for onion over a 15 month period. The wide price difference is due to a combination of the aforementioned noise and economy volatility. The multi-modal shape of the distribution is also noteworthy, where multiple different prices were frequently quoted for a single food commodity such as onion.

## RESULTS

### The nowcast model

The challenge in determining a representative daily price trajectory from thousands to millions of price quotations on social streams is handling noise. This is because the raw price quotations span a wide price range and show muti-modal distribution, as shown in the example case of onion in Fig. 1B. Utilizing the raw tweet data without any screening of extremely high or low price values results in poor price prediction for two primary reasons. First, the predicted price from raw tweets could have disproportionately large spikes. For example, the beef price surged 17.5 times compared to the official price for certain days in July 2012 based on our tweet data, which should be considered as outliers. Second, such outliers lead to an overall poor quality of price prediction measured by the mean absolute percentage of error. Simply eliminating outliers would yield a large reduction in prediction error. Therefore, devising a filter to eliminate unnecessary noise and find meaningful signals from the dataset is critical for price prediction.

We propose a new nowcast model that is suitable for accommodating food price dynamics. The proposed nowcast model is depicted in Fig. 2, which takes in raw price quotations from social media streams as input and outputs a single price value per day for each commodity. Noise in the dataset is determined by examining the discrepancy between today's price quotations against yesterday's official price. In the model we assume market prices are non-stationary time series; this is consistent with the assumption that has been made in relevant studies (*Leuthold, 1972*; *Working, 1934*). We further consider the Markov process for price dynamics as assumed in *Zhang (2004)* and *Ghasemi et al., (2007)*. Hence,

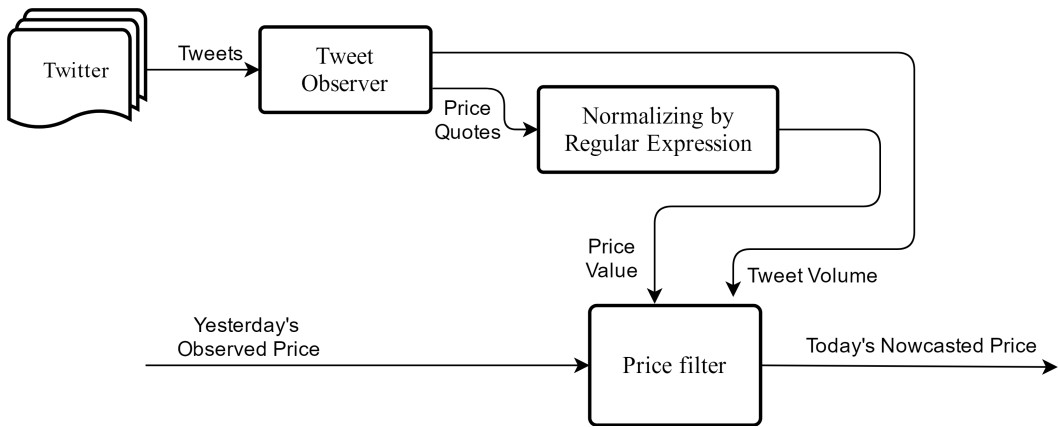

**Figure 2** **Framework of the nowcast model.** The model takes in price quotations from social media streams and predicts today's commodity price via jointly considering yesterday's price with today's price quotations.

let today's price $P_t$ be determined both by yesterday's price $P_{t-1}$ as well as today's price quotations from Twitter $P_t^{\text{tweet}}$. The weighting factors in the Eq. (1), $\alpha$ and $\beta$, represent the relative importance of these two quantities on today's price. The model would then respond to the current market quotes faster when $\beta$ is larger than $\alpha$, in which case a larger degree of price fluctuations are expected.

$$P_t = \frac{\alpha P_{t-1} + \beta P_t^{\text{tweet}}}{\alpha + \beta}. \tag{1}$$

Furthermore, we assume that daily food prices do not change radically. The maximum change in commodity price that we observe from historical data is marginal for most days. For instance, the largest deviation seen for the beef price was changing by 2.5% from one day to another on Aug 16th 2012. This observation leads us to assume that prices of a commodity on a given day and the consecutive day would be within certain bounds. This is modeled as a variable $\delta$ defining the maximum allowable price change rate. Any social signals that exceed this change limit from one day to another will be eliminated from analysis at the outset. Hence if a quoted tweet exceeds this threshold compared to the previous day, the model rejects it as a valid input. Equation (2) describes this constraint, where $T_t^i$ is an $i$th individual tweet price which is taken from day $t$.

$$\text{if } \left| \frac{T_t^i - P_t}{P_t} \right| > \delta \text{ then eliminate } T_t^i. \tag{2}$$

Another assumption is made for calibrating the effect of tweet volume. Twitter signals are generated significantly more on days where the price change is larger, as shown in Fig. 3. Based on this finding, the logarithmic value of tweet volume was used as the weighting parameter $\beta$ in order to give disproportionately higher impact on days with large social signal. In case there was no social signal (i.e., zero tweet), the nowcast model assumes there is no change in price. On the other hand, in cases when food commodity prices decrease, people may tweet the price less frequently. To accommodate such data scarcity problem, the proposed nowcast model refreshes when there is no tweet for $n$ consecutive days. The

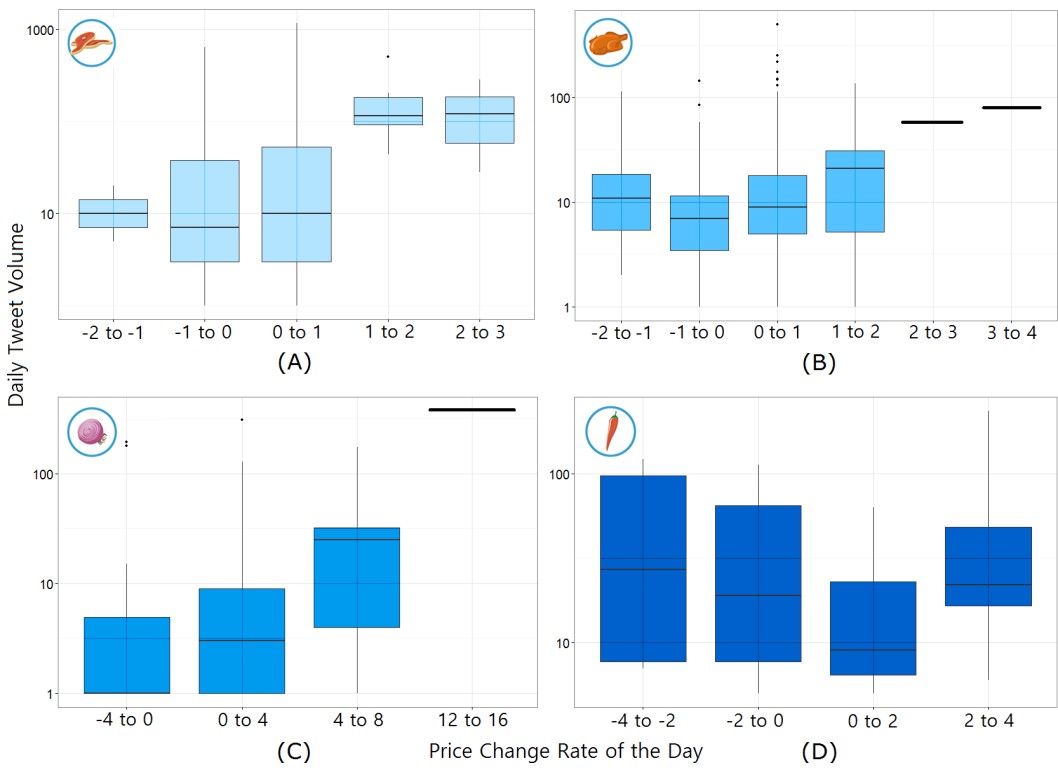

**Figure 3  Daily tweet volume according to price change rate of the day for each commodity.** (A) Beef, (B) Chicken, (C) Onion, and (D) Chilli. It shows a tendency for more people to talk about food prices when they goes up or down; however, the number of daily tweets itself cannot be a predictor.

model takes the average price estimates from the recent $k(k \gg n)$ days. We demonstrate this example in Article S1. The main idea is to restart the model with a starting price of the recent average price (from $k$ days before today) since the model price cannot be guaranteed after any zero-tweet period.

Equation (3) shows the final model with four parameters: $\alpha$ (the ratio between the weights of yesterday's price and today's tweet price), $\delta$ (the allowed maximum daily price change rate), $n$ (the number of zero-tweet dates for restarting computation), and $k$ (the period over which the average commodity price is calculated). $Q_t^j$ refers to the individual price quotation from tweets, while $[Q_t]$ is the number of daily tweets. We set the starting price $P_0$ as the commodity price on the first observation date.

$$P_t = \frac{\alpha P_{t-1} + log([Q_t]+1)P_t^{tweets}}{\alpha + log([Q_t]+1)}$$

$$P_t^{tweet} = \frac{\sum_{j=1}^{[Q_t]} w_t^j Q_t^j}{\sum_j w_t^j} \qquad w_t^j = \begin{cases} 1 - \dfrac{\left|\dfrac{Q_t^j - P_{t-1}}{P_{t-1}}\right|}{\delta} & , if \left|\dfrac{Q_t^j - P_{t-1}}{P_{t-1}}\right| \leq \delta \\ 0 & , otherwise \end{cases}$$

$$P_{t-1} = \frac{\sum_{j=t-k}^{t-1} P_j}{k} \text{ where no tweets over } n \text{ days.} \tag{3}$$

## Existing price prediction models

Previous studies have proposed several different models of price prediction that can be used in the context of social media price quotations. The first model we review is the inter-quartile range (IQR) filter model that eliminates any extremely low or high price quotations and accepts prices between the upper and lower quartile on a given day. The IQR filter is useful, when a distribution has central tendency and when the majority of data is placed nearby to form a truthful range. While this is a simple model, the IQR is known to perform poorly when the data have a distribution of multiple peaks, as in the case of the price quotations we observe on Twitter.

Second, density estimation models such as the kernal density estimation (KDE) are effective for single-dimensional multi-modal data, which are typical cases in price data as seen in Fig. 1B. The KDE algorithm is a non-parametric method that estimates the probability density function of a random variable. Local minima in the density function from KDE can be used as a split point of data into clusters, thereby allowing one to identify the largest cluster of daily price quotes. The largest cluster on any given day indicates price values that are the most commonly quoted and hence can be considered as the most credible prices. We set the bandwidth of the kernal function by minimizing the mean absolute percentage of error (MAPE) with 80% of the randomly-chosen tweets over the first three months.

A third model considered is the auto-regressive integrated moving average (ARIMA), which is a widely used approach for forecasting trends in time series data. ARIMA model is a generalization of the Auto-Regressive (AR) model that predicts output values by its own previous values. The parameters of the ARIMA model were determined by the corrected Akaike information criterion (AICc) values (*Hyndman & Khandakar, 2007*) based on the first three-months worth of the official price data.

A fourth model is the linear model proposed for the Google flu trend, which adopts a linear regression function on logit space, where $I(t)$ is the predicted influenza rates at time $t$, $Q(t)$ is the influenza-related query fraction at time $t$, $\alpha$ is the multiplicative coefficient, $\varepsilon$ is the zero-centered noise, and $\beta$ is the intercept term: $logit(I_t) = \beta + \alpha \cdot logit(Q_t) + \varepsilon$. However, this model cannot be directly applied on Twitter for several reasons. One reason is that the linear correlation between tweet frequency and price change is not strong (Pearson correlation $r = 0.17$, $p < 0.01$) and in fact we find support for non-linearity. Another reason is that commodity price quotations on Twitter are sparsely distributed in time (e.g., zero-tweet days) compared to the rich source of data such as the Google search query. For these reasons, we do not directly compare our results with the Google flu trend-like model.

## Prediction performance

Prediction performance of the nowcast model is measured and compared to existing models in two ways: (1) trend forecasting via the Pearson correlation coefficient $r$ and (2) error rates via the mean absolute percentage of error (MAPE) between the official and estimated prices. Some parameters in the model are independent of the intrinsic properties of food commodities. For instance, the relative responsiveness of the model to yesterday's price

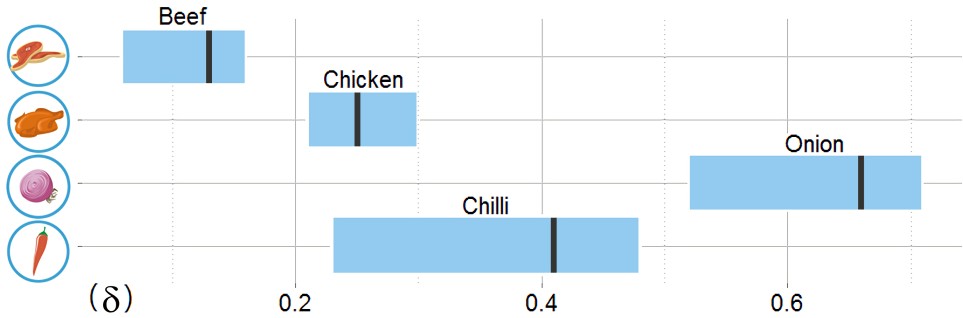

**Figure 4** **The allowable model parameter δ ranges for four target food commodities based on training data.** All allowable delta ranges include four times the historical maximum daily price change rate, which are displayed with a vertical line for each commodity.

($\alpha$) and the thresholds to restart the model after a period of infrequent tweets ($n$ and $k$) are assumed in the model and hence are set as follows: $\alpha = \log(21)$, $n = 7$ days, and $k = 60$ days. Other parameters, in contrast, were tuned to best describe the data. For instance, the maximum daily price change rate ($\delta$) is trained separately for each food commodity and the starting price at day 0 of prediction ($P_0$) is set separately for each commodity as the commodity price on the first observation date (June 1st, 2012).

In determining $\delta$, a parameter that determines which tweets are accepted or ignored in the model, we examine the price change dynamics from historical records. The beef price changed gradually with a maximum price change of no more than 2.5% from one day to the next, whereas onion showed a rapid change in price with a maximum change rate of 15.1% from one day to another. This means that the daily allowable change rate should be set higher for onion compared to beef. We set $\delta$ by training with a randomly-chosen 80% of the first three-months of tweets, which are identical to the training set for other comparison models, so that the nowcast model correlation $r$ exceeds 0.80 and RMSE is within 10% of each commodity price. The allowable range of $\delta$ are shown in Fig. 4. Performance variation in terms of $r$ according to change of $\delta$ across all target commodities is shown in the (Fig. S1). For further evaluations, we use the identical test set for all models, which is the remaining 12 months of data after excluding the training period.

Next we examine the prediction performance via the percentage of error of the daily prediction, measured by taking the difference between the official and estimated price divided by the official price. Figure 5 shows the distribution of the percentage error for all four commodities over 12 months; the minimum, 25th percentile, the median, the 75th percentile, and the maximum error ranges are shown for the nowcast model as well as three existing models, the IQR model, the KDE model, and the ARIMA model. The time series based models (i.e., ARIMA and Nowcast) perform better than the statistical filter-based models (i.e., KDE and IQR) given by the shorter error ranges. The memory structure of the time series-based models and their regressive correcting process may contribute to better fitting results for ARIMA and Nowcast. Between these two models, the median percentage error of Nowcast is consistently smaller.

Table 3 shows the result for both the correlation and the absolute error. Again, IQR and KDE do not yield the same level of performance as time series-based models. ARIMA yields

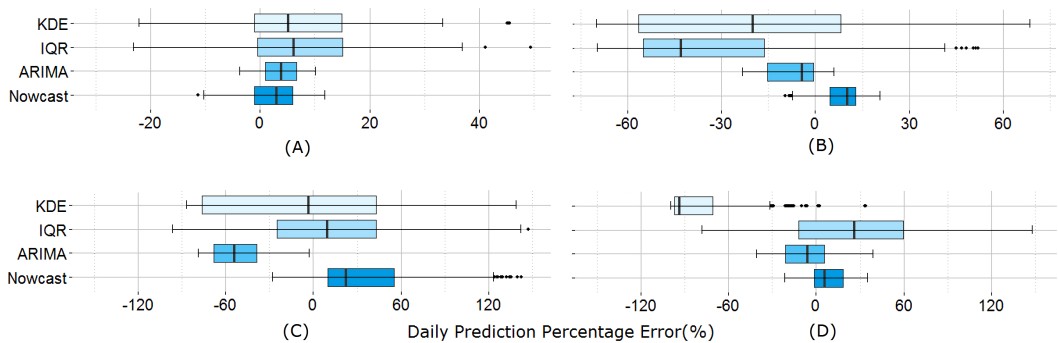

**Figure 5** **Daily prediction error comparison between the models.** (A) Beef, (B) Chicken, (C) Onion, and (D) Chilli. Time series based prediction models (ARIMA and Nowcast) show better performance in terms of error range.

**Table 3** Prediction performance comparison between the models.

| Commodity | Total tweets | NOWCAST | | ARIMA | | IQR | | KDE | |
|---|---|---|---|---|---|---|---|---|---|
| | | $r$ | MAPE(%) | $r$ | MAPE(%) | $r$ | MAPE(%) | $r$ | MAPE(%) |
| Beef | 14,473 | 0.85 | 4.86 | 0.60 | 5.02 | 0.29 | 18.05 | 0.25 | 11.14 |
| Chicken | 5,223 | 0.84 | 9.41 | 0.42 | 8.74 | 0.46 | 46.45 | 0.34 | 45.87 |
| Onion | 1,954 | 0.85 | 32.79 | 0.35 | 42.88 | 0.60 | 40.83 | 0.63 | 43.36 |
| Chilli | 1,772 | 0.76 | 13.62 | 0.51 | 11.26 | 0.32 | 70.21 | −0.25 | 81.35 |

the smallest MAPE for certain commodities like chilli and chicken, yet the correlation coefficient ($r$) remains the highest for Nowcast. This may be due to the non-stationary property of the price trend data in developing regions, which is handled better by the proposed nowcast model. Economic time-series are commonly far from stationary in their original form, and ARIMA requires stationarizing process by de-trending or differencing. However, we find that price time-series of Indonesian commodity price is neither a trend-stationary nor a difference-stationary. Unlike ARIMA, the Nowcast model does not rely on a pre-assumption of stationary price time-series, which is hardly applicable for developing countries' volatile economic status. For monitoring economic markets, the ability to represent trend dynamics is as important as reducing the absolute error. Hence this comparison demonstrates that the nowcast model outperforms existing models.

## Time-lagged correlation

Beyond investigating the raw correlation in data, we test whether adding any time lag would better explain the relationship between the official and predicted prices. We utilize the cross-correlation coefficient (CCF) to estimate over what time lag the two price time series data are related. The CCF value at lag $\tau$ between two time series data measures the correlation of the first series with respect to the second series shifted by $\tau$ days (*Ruiz et al., 2012*). For each target commodity, Table 4 displays that there are maximum positive correlations at lag of 0 or +1 day, meaning that the model has the highest accuracy within a single day lag. According to the literature, nowcasting is defined as the capability to capture information on a real-time basis within a short time gap typically in the single day range

**Table 4  Cross correlation between official and nowcasted prices across target commodities.**

| Commodity | Lag (days) | | | | | | |
|---|---|---|---|---|---|---|---|
| | −3 | −2 | −1 | 0 | +1 | +2 | +3 |
| Beef | 0.28 | 0.19 | 0.62 | 0.85 | 0.79 | 0.50 | 0.41 |
| Chicken | 0.29 | 0.24 | 0.77 | 0.84 | 0.63 | 0.42 | 0.33 |
| Onion | −0.13 | 0.32 | 0.68 | 0.85 | 0.83 | 0.67 | 0.13 |
| Chilli | 0.41 | 0.09 | 0.49 | 0.76 | 0.81 | 0.31 | −0.20 |

(*Giannone, Reichlin & Small, 2008*). Hence, we can conclude that the suggested model is capable of nowcasting daily food prices in Indonesia. Table 4 also indicates that there are the highest positive correlations at lag 0 to +1 for all commodities, meaning that a daily price value nowcasted from social media has a predictive power on the price value of the next day.

## DISCUSSION

This study shares insights into building an affordable and efficient platform to complement offline surveys on food price monitoring. The market data gathered through social media help to predict economic signals and assist food security decisions. Price quotations in social media are a new type of information that need extensive cleaning before usage. A naive statistical filtering method is no longer effective, because price distribution is not normally distributed and contains various noise elements as shown in Fig. 1B. The proposed nowcast model attains acceptable performance with a simple filtering method that does not rely on sophisticated natural language processing techniques. In applying the suggested model to other languages, a taxonomy of keywords related to commodity names and prices would need to be identified. Our model has minimum language dependency and no grammatical considerations are required. Its filter operates via keyword extraction and numerical analysis based on the characteristics of the Twitter data. The model can also handle data sparsity, this quality is important given that people do not always mention prices on social media.

The nowcast model, which is tested successfully on four main food commodities in Indonesia, can be adapted to predict trends in other essential commodities and across countries. Our evaluation proves the accuracy of the nowcast model by comparing prices extracted from public tweets with official market prices. The tool, hence, could operate as an early warning system for monitoring unexpected price spikes at low cost, complementing traditional methods. Therefore, this work has implications in terms of demonstrating a simple and replicable technical methodology—keyword taxonomy refined by numerical filters—that allows for straightforward operational implementation and scaling.

### Social network-wide sensitivity to price fluctuations

The premise of this paper lies in the assumption that social network users such as those on Twitter not only voluntarily share information about food prices but also these signals are sensitive enough to capture day-to-day price fluctuations. If there are not enough tweets mentioning food prices, algorithms like nowcast will face a data scarcity problem. In fact,

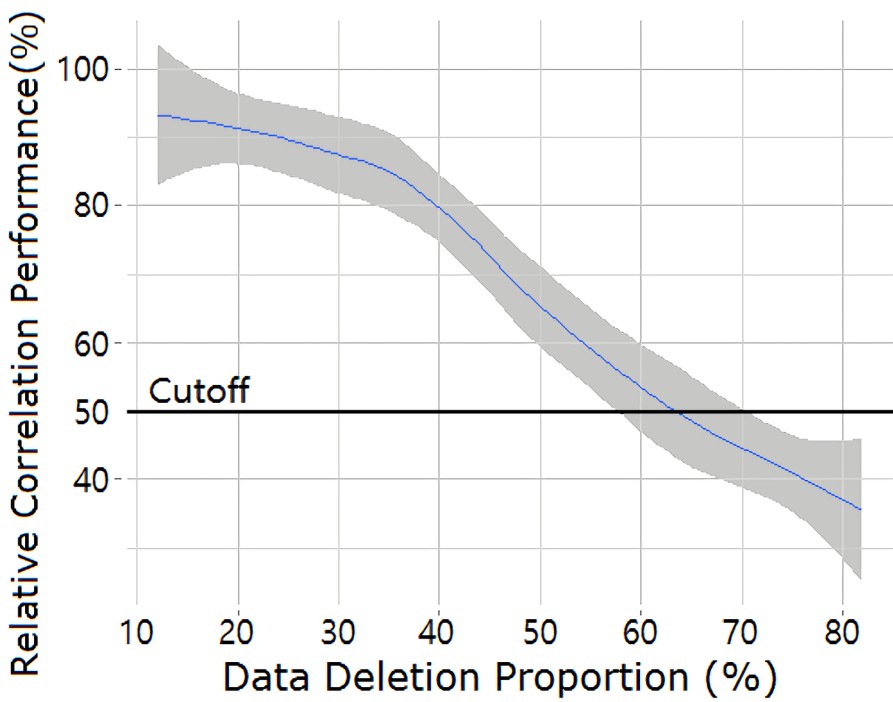

**Figure 6** **Decaying of the relative correlation performance ($r$) as a function of the scale of data removal for the most scarce commodity, chilli.** Each data point indicates the average performance of 50 runs after randomly choosing to remove a given fraction of price quotations. The blue line indicates the relative increment/decrement (%) of the averaged correlation against the non-removing case and the shaded area represents the ranges of outcome across all 50 trials.

data shortage can be witnessed in the historical data. Tweets that mention food prices occupy no more than 0.07% of the entire tweet dataset in Indonesia and users on average post no more than a few tweets a year on such a topic (2.7 tweets over 15 months).

Here we check the robustness of the algorithm under extreme challenges involving noise and lack of data with the least mentioned commodity, chilli. Out of the entire 484-day observation period, chilli was not mentioned once over 312 days and fewer than three times over 87 days. To test the robustness of the nowcast algorithm under data scarcity, a random set of chilli-related tweets accounting 10% to 80% of total are removed and the price is predicted with only the remaining data. For each simulation, data elimination is repeated 50 times and the averaged performance results are reported for comparison. Figure 6 shows the prediction quality $r$ (Pearson's correlation) as a function of the data deletion ratio. We find the trend forecasting to remain relatively stable until a moderate level of data deletion; the $r$ value is degraded no more than 20% until 40% of data is eliminated. The $r$ value starts to decrease more rapidly after this point although still reaching a correlation of above 50% until 65% of data is eliminated. This high resilience to noise for the case of chilli demonstrates that the nowcast model can handle well the level of data scarcity seen in real data. Other food commodities, which are more frequently mentioned, show an even higher level of resilience to noise.

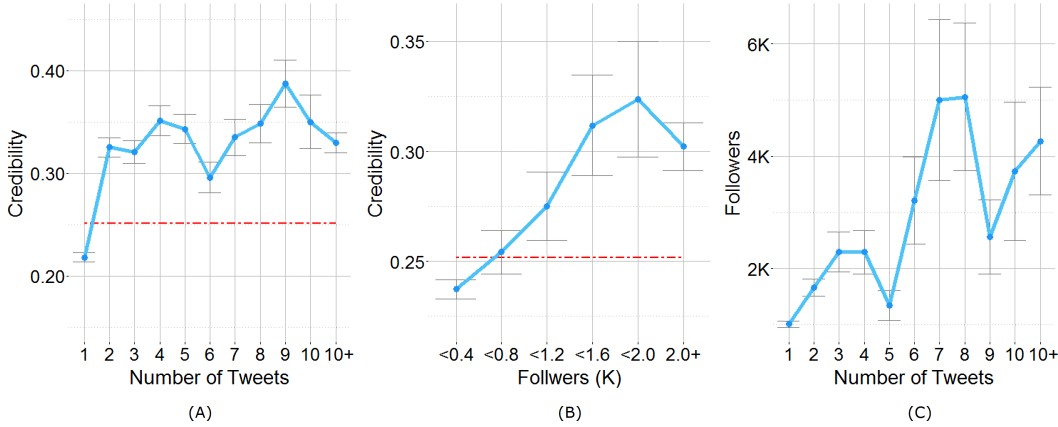

**Figure 7** (A) Users' credibility plot versus their number of Tweets. The dashed red line depicts the mean credibility of all users (=0.251) (B) Users' credibility plot versus their number of followers. (C) Users' followers versus their number of Tweets.

Another issue to be considered is the nowcast model's sensitivity to price fluctuations. We find that the model achieves better predictive power under large price variations; there exists a negative correlation between the daily price increase rate and model error ($r = -0.52$). This might be explained by several causal factors. For instance, the volume of price quotations is affected by how the actual food price changes; people tend to post more tweets during periods of price inflation than price deflation (Fig. 3). This tendency is more apparent on food commodities that often experience volatile price fluctuations. For instance, onion receives on average 11.3 times more tweets upon price inflation than price deflation. Tweet volume is directly related to the richness of the data source for the nowcast model, and hence its performance depends on price trends. The partial correlation between the price change rate and the model error after controlling for tweet volume is considerably lower ($r = -0.27$).

## Credible users

While the nowcast model treats individuals on Twitter equally and utilizes all tweets that are within the allowed price ranges, one may look further into whether a smaller set of highly credible users exist and if so what their common traits might be. *An & Weber (2015)* have shown that different user-level sampling strategies can affect the performance of nowcasting on common offline indexes. Based on their work, we test whether accounts that quote prices more frequently in fact mention more accurate prices. We define the credibility of an account and examine its relationship with tweet volume. Credibility is defined as the ratio of credible tweets over entire quotations posted by the same user, where credible tweets indicate those tweets picked by the model in allowable price range (i.e., the mentioned price is within the $\delta$ range from the predicted price of the previous day).

Figure 7A displays user credibility, grouped by the number of price quotations during the observation period. Overall, Twitter users in Indonesia had an average credibility of 0.252, indicating one out of four tweets could be used for price prediction in the nowcast model. Those who quote food prices more than one time have 1.2–1.5 times higher credibility

scores than the average. Nonetheless, there is no significant correlation between the tweet volume and credibility at the user level (Spearman correlation coefficient of 0.048), indicating that accounts that mention prices frequently are not necessarily credible. In particular, the top-ten most prolific accounts are food vendors, who send out provocative advertisements that may not represent the real commodity price.

Another measure we consider is user degree or the follower count. Social media comprise users of various influence levels, which can be measured by metrics like the user degree. Would influential users generate more credible tweets when it comes to food prices? Users who mention food prices have far more followers than the average. The mean degree in the studied Twitter network is 1,422 with a median of 220, which indicates a one-fold difference compared with what had been reported in other Twitter studies (*Cha et al., 2010*). The correlation between user degree and credibility is also significant (Spearman $r = 0.320$), indicating that accounts with more followers mentioned more accurate food prices (Fig. 7B). Furthermore, those who tweeted food prices more frequently tend to have more followers (Spearman $r = 0.183$) as shown in Fig. 7C. These observations lead us to conclude that while there is no direct correlation between the level of credibility and tweet volume, having more followers leads to a positive effect on quoting credible food prices. While the current nowcast model does not consider any user traits, it may be interesting to explore the idea of finding more informative and influential user groups for economic indicators.

## Summary

The proposed nowcast model shows remarkable potential in tracking daily food commodity prices with high accuracy in the case of Indonesia, where official statistics on food are, at times, gained with a delay of several days. Given the volatile nature of the economy in developing countries and their resource hungry monitoring systems, online big data help address the limitations of traditional official statistics by allowing fine-grained prediction of economic trends (*Ruiz et al., 2012*). Government actions that lead to temporal fluctuations of food prices are common in developing countries. For instance, the Indonesian government occasionally imports meats and other farm products to stabilize food prices. Governments sometimes also donates seeds to farmers or sell them at lower prices in the hope of increasing supply from the next harvesting season (*Sambijantoro, 2015*; *CustomsToday, 2016*). With faster monitoring of financial fluctuations, governments in developing economies can make better policy decisions to protect vulnerable populations. The nowcast model can predict daily food prices through a longitudinal period of 15 months, as demonstrated in Fig. 8.

Traditional statistics and surveys nonetheless remain a practical and accurate source of information for establishing the ground truth. The presence of online big data complements the official data by providing transient views. From this perspective, the nowcast model acts as a supporting tool for official statistics than as a stand-alone system. In particular, nowcasting will be more valuable for short-term forecasts before releasing official statistics, as mentioned by the Organization for Economic Cooperation and Development (OECD) and the United Nations Statistics Division (*United Nations Leadership Council of the Sustainable Development Solutions Network, 2015*; *Schiefer, 2012*). Future work will need

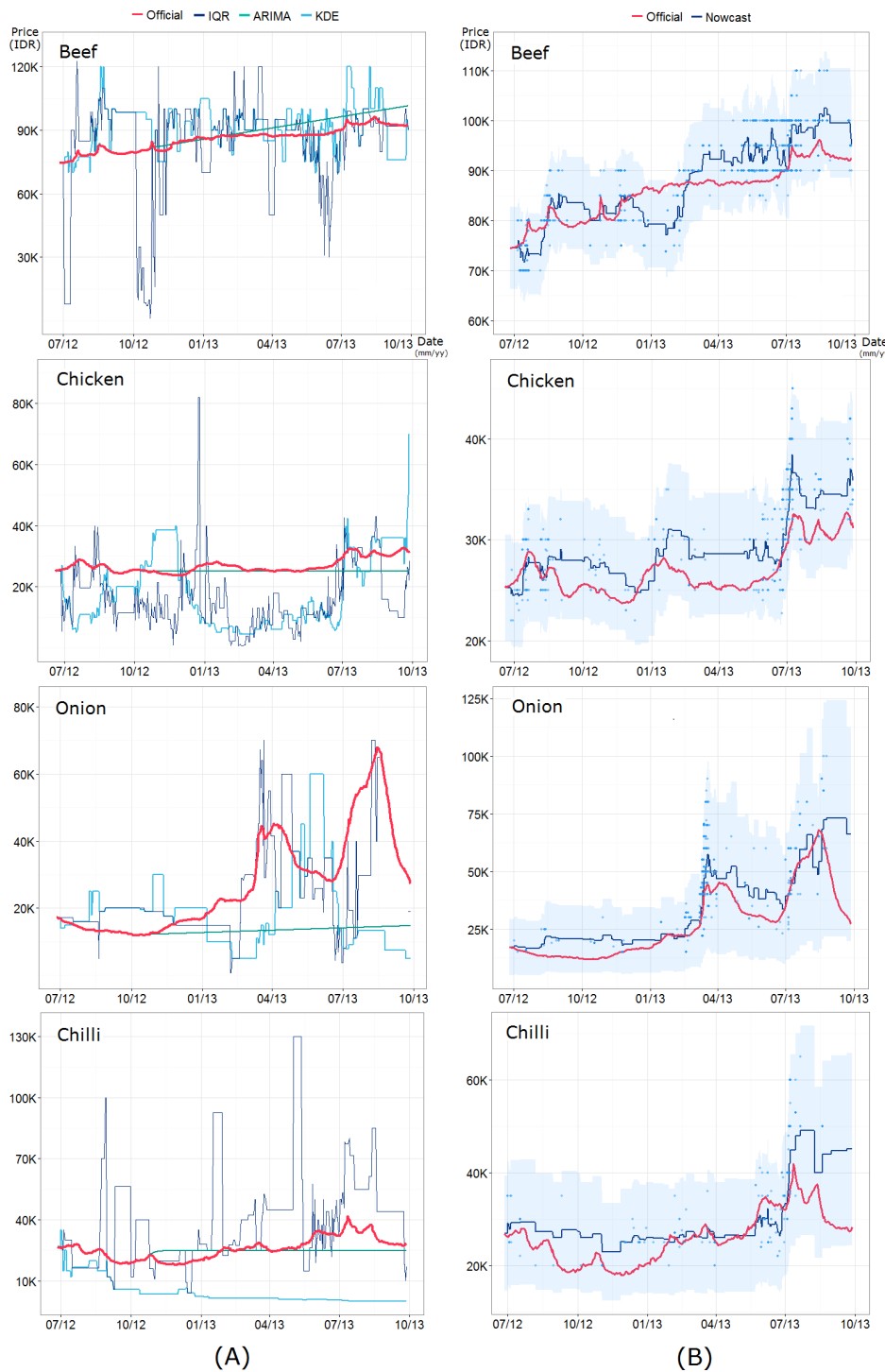

**Figure 8** **Full comparison of (A) the three alternative models—interquartile range (IQR) filtering, ARIMA model, and kernel density estimation (KDE) clustering—and (B) the proposed nowcast model across four food commodities.** The blue points indicate the price quotations from Twitter and the shaded area represents the credible price range determined by a model parameter δ.

to focus on how to combine traditional market surveys and social media-based nowcast to maximize their predictive performance. Performance of the nowcast model still can be improved by using of additional datasets such as Google Trends and other Web search records. Such extension might be particularly useful in developing countries that have far fewer social media users. We further expect that periodical feedback from official statistics can make the model more practical to provide early warning of unexpected price spikes at a lower cost than traditional data collection.

### Funding

This work was supported by the Ministry of Trade, Industry & Energy (MOTIE, Korea) under Industrial Technology Innovation Program (No.10073144), 'Developing machine intelligence based conversation system that detects situations and responds to human emotions'. There was no additional external funding received for this study. The funders had no role in study design, data collection and analysis, decision to publish, or preparation of the manuscript.

### Grant Disclosures

The following grant information was disclosed by the authors:
Industrial Technology Innovation Program: 10073144.
Developing machine intelligence based conversation system.

### Competing Interests

Meeyoung Cha is an Academic Editor for PeerJ.

### Author Contributions

- Jaewoo Kim performed the experiments, analyzed the data, contributed reagents/materials/analysis tools, wrote the paper, prepared figures and/or tables, performed the computation work, reviewed drafts of the paper.
- Meeyoung Cha conceived and designed the experiments, contributed reagents/materials/analysis tools, wrote the paper, reviewed drafts of the paper.
- Jong Gun Lee analyzed the data, contributed reagents/materials/analysis tools, wrote the paper, reviewed drafts of the paper, data sourcing.

### Data Availability

Indonesian tweet data of commodity price quote:
dx.doi.org/10.7910/DVN/XWM9VB.

### Supplemental Information

Supplemental information for this article can be found online at http://dx.doi.org/10.7717/peerj-cs.126#supplemental-information.

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
