# Peer review of "Nowcasting commodity prices using social media"

_PeerJ Computer Science, doi:10.7717/peerj-cs.126_

## Round 0.1 · original submission · Major Revisions

This is a very interesting paper -- predicting food commodity price is a very relevant social-economic problem, and this paper is the first (that I know of) to tackle this problem by overcoming/implementing a number of technical solutions.

I strongly encourage the authors to take into account comments from both reviewers, including but are not limited to: describing data sources for ground truth, evaluation protocol, articulate example applications (e.g. gov make purchasing decisions based on expected price fluctuation), additional references, robustness of NLP techniques, and writing improvements.

Granger causality is a somewhat strange choice of measuring prediction quality since it measures the quality of x(1:t-1) vs y(t) but not x(t) against y(t). The error metric MAPE makes more sense. Perhaps the authors can also consider correlation metrics (e.g. Pearson).

The choice of prediction target also seem under-explored in this article. An earlier work on twitter --> dow jones prediction (https://arxiv.org/pdf/1010.3003.pdf) the target was decided as the direction of change (up/down) rather than the absolute value. In this paper the predicted price is a moving-average between historical price and a heuristic combination of tweet volume and price mention in tweets. I wonder (1) if the price prediction target was set to up/down and then a few discrete buckets of change (e.g. minor, small, large) the predictor will do well. (2) If the predictor shall improve using standard machine-learning approaches, such as logistic/linear regression on a small set of features — intuitively it would.

I think it will be very interesting to see explanation of the model, e.g. representative tweet and the prediction, and a brief discussion about when/why things fail sometime (aside from missing data).

Supplemental material shows sufficient care that the authors have taken to ensure the quality of data and present complete results. In my opinion they are an integral part of the paper and will be useful for someone who want to reproduce the system. Assuming there's no PeerJ page limit, I suggest that they are included in the main text, or after references, rather than as separate files. If this is not possible, then a short list of what the appendixes are should be included in the main text likely towards the the end.

I have put a major revision rating due to the additional comments above, and a desire to see comments from both reviewers thoroughly addressed and potentially re-checked by reviewers.

·

Basic reporting

The authors are commended for reporting in a largely clear and unambiguous manner, particularly across languages. However, the English language should be improved through a closer proof-reading. Some examples include:

- Lines 31-32: confusing/disjointed sentence (e.g., break into two sentences);
- Footnote 1: pluralisation - "region" should be "regions";
- Line 70: check phrasing - "Indonesia is one of most tweeting countries..."
- Line 94: pluralisation - "is" should be "are";
- Line 108: missing word - "...we suggest [a] Nowcast model...";
- Line 113: check phrasing - "...from June 2012 to Sep 2013 total 78,518 tweets from 28,800 accounts..." --> e.g., could be rephrased as: "... to Sep 2013 [a] total [of] 78,518 tweets...";
- Line 136: "to be" --> should be "was";
- Figure 1 caption: "...variations due [to] geographic diversity..." (missing word). Also "multi-model" should be "multi-modal"?;
- Line 202: "The fist model we review..." --> should be "The [first] model we review...".

Terms and theorems are reasonably clear. However, there is some unnecessary repetition. For example, the variable δ (delta), which assigns bounds on price rate change, is defined several times (Lines 174-176; Lines 188-189; Lines 194-195).

Although sufficient background/context is provided, I think it would greatly enhance the paper if the authors provided more information about significance of the domain of the study. Food shortages in developing countries constitutes a serious social and governmental issue, and this research provides a valuable contribution to solving such problems. For example, in the Summary section, the authors state that "governments ... can make better policy decisions to protect vulnerable populations" (Lines 378-379). A brief sentence or two about how this research can enable "better policy decisions" would add value to the paper, particularly for interdisciplinary readers in policy and administration.

Experimental design

By and large, the experimental design is sound and to be commended. However, there are two minor aspects that would need to be addressed (or at least clarified) prior to accepting the paper.

Firstly, the aims and research questions and/or hypotheses of the study are not clearly defined. With some effort it is possible for the reader to discern these (e.g., from the Introduction section). I recommend that the authors (1) devote a brief paragraph to clearly state the aim(s) of the study; and (2) devise a set of research questions and/or hypotheses that clearly define what the study seeks to answer. These could be presented in a brief table on Page 2, for example. For the latter, this is particularly useful in relation to comparing the performance of alternative models and also for the analysis of Granger causality.

Secondly, the authors should be commended for their novel approach to filtering and analysing socially-generated data that are, by nature, noisy and error-prone. However, I query the removal of bot accounts (see Lines 123-126) from the data. The rationale for removing bots is that these accounts "have a disproportionate impact on the price monitor". However, the literature tells us that 'socialbots' play an important role in, for example, shaping markets and financial events (Steiner, 2012) and politics (Graham and Ackland, 2016; Ratkiewicz et al., 2011). In this way, the bots may have a potentially important 'social' role to play in commodity prices in Indonesia, even if their presence complicates the modelling problem. If the authors could clarify the removal of bots, that would be helpful.

References cited in Section 2 (of this review):

Graham, T., & Ackland, R. (2016). Do Socialbots Dream of Popping the Filter Bubble? The role of socialbots in promoting participatory democracy in social media. In Bakardjieva, M. & Gehl, R. (Eds.) (2016). Socialbots: digital media and the automation of sociality. Routledge: New York.

Ratkiewicz, J. Conniver, M. Meiss, M. Gonçalves, B. Patil, S., Flamini, A., and Menczer, F. (2011). Truthy: mapping the spread of astroturf in microblog streams. In Proc. 20th ACM Intl. World Wide Web Conf. Companion (WWW), pp. 249–252.

Steiner, C. (2012). Automate this: How algorithms came to rule our world. New York: Portfolio/Penguin.

Validity of the findings

Overall the findings are interesting and valid. I have two relatively minor comments for further improvements to the paper.

Firstly, the application of Granger causality (Lines 268-289) augments the analysis nicely and adds depth to the results. However, I have a point of clarification. The authors assume that, in this study, market prices are a non-stationary time series. This is fine. However, given that Granger causality assumes the signals to be covariance stationary, could the authors please outline any appropriate transformations they applied to the non-stationary data, i.e., to transform the time series into a stationary process. This will help to make the research reproducible and clarify the statistical rigour.

Secondly, as per Section 1 of this review, it would be helpful to clearly define the research questions at the beginning of the paper, in order to link these to the findings and conclusion at the end of the paper. I believe the findings to be valuable, both empirically and theoretically, so linking these clearly to research questions will only enhance this research.

Additional comments

Overall, the authors are commended for contributing a novel technique for modelling market prices based on social media data, as well as eliciting interesting empirical results to inform public policy and future research. I also praise the applied aspect of this paper (improving public policy in developing countries). I recommended this paper for publication, subject to a set of minor revisions, as outlined in detail in this review. Best wishes to the authors.

Reviewer 2 ·

Basic reporting

This article describes the use of prices reported in social media data to predict commodity prices. Focusing on four food items in Indonesia, the paper presents the modelling approach and evaluation results comparing the prediction to ground truth data. The paper also describes additional exploratory analyses to assess: (1) effect of data volume; and (2) credibility and user modelling.

Overall this article was a very insightful read. The use of social media analytics to commodity prices is an interesting application. The paper suffers at times from grammatical errors (see below for minor edits), however the overall presentation of the information and argument are clear and easy to follow.

Experimental design

In terms of the evaluation of the proposed model, the article could be strengthened by clarifying (1) the nature of the gold standard data; and (2) the usage of the data.

1. In this work, the authors describe having access to historical data for actual food prices. The source should be described and referenced if possible. Some metadata description about the historical data would also be useful for the reader. What time granularity does this come in? How is the historical data collected? How easily is this data compiled (perhaps to motivate the use of social media data).

2. More importantly, the authors describe the use of the historical data to determine values for the parameters to the model, such as delta, the maximum allowable price change rate (line 195). The authors describe this value as being tuned based on 80% of the data (line 245). Presumably, the results reported in Figure 5 and Table 2 are on held-out data (presumably 20%). However, this isn’t stated clearly anywhere. Needless to say, if this is not the case (the reported results is on the whole historical data set), the NOWCAST model has some unfair advantage over the other models.

Validity of the findings

If the authors have used held-out data for the evaluation, the comparisons and findings seem sound (see previous section on Experimental Design).

Additional comments

This article describes the use of prices reported in social media data to predict commodity prices. Focusing on four food items in Indonesia, the paper presents the modelling approach and evaluation results comparing the prediction to ground truth data. The paper also describes additional exploratory analyses to assess: (1) effect of data volume; and (2) credibility and user modelling.

Overall this article was a very insightful read. The use of social media analytics to commodity prices is an interesting application. The paper suffers at times from grammatical errors (see below for minor edits), however the overall presentation of the information and argument are clear and easy to follow.

***Improvements***

In terms of the evaluation of the proposed model, the article could be strengthened by clarifying (1) the nature of the gold standard data; and (2) the usage of the data.

1. In this work, the authors describe having access to historical data for actual food prices. The source should be described and referenced if possible. Some metadata description about the historical data would also be useful for the reader. What time granularity does this come in? How is the historical data collected? How easily is this data compiled (perhaps to motivate the use of social media data).

2. More importantly, the authors describe the use of the historical data to determine values for the parameters to the model, such as delta, the maximum allowable price change rate (line 195). The authors describe this value as being tuned based on 80% of the data (line 245). Presumably, the results reported in Figure 5 and Table 2 are on held-out data (presumably 20%). However, this isn’t stated clearly anywhere. Needless to say, if this is not the case (the reported results is on the whole historical data set), the NOWCAST model has some unfair advantage over the other models.

***General comments and questions***

3. Related work
At a high level, it might be interesting to comment on other well-known uses of Twitter for prediction. For example, the work by Asur and Huberman (2010) in using Twitter to predict movie box office sales is rather well known (http://dl.acm.org/citation.cfm?id=1914092). It would be interesting to know how the proposed model compares to prior work such as the one proposed by Asur and Huberman. Even a discussion about the applicability of the prior work would be useful.

4. With respect to Natural Language Processing
The decision to use minimal natural language processing techniques is understandable. However the article presents the model as being robust to variations in the way one can express prices, leaving it to the model to make the appropriate inference.

Line 107: “Instead of using hundreds of regular expressions for normalizing 108 various types of units into an identical units, we suggest Nowcast model which can handle a commodity 109 unit difference issue via numerical approach.”

This is perhaps due only to the specific commodities analysed in the article. (One would need more evidence and examples to show that text variation is systematically handled by the model, which seems unlikely.) Perhaps, state that, for the purpose of simplicity of the model, by assuming that prices for specific foods fall within distinct price ranges.

Line 297, “Hence, our model has minimum language dependency”. This should be clarified to say that aside from the taxonomy (Table 1) and the queries used to collect the data, the approach makes little use of linguistic information. Imagine the case for Chinese, Korean or Japanese where word segmentation is a problem and may hinder the creation of similar regular expressions as found in Table 1.

5. With respect to data/results

On Line 96, the authors write: “Four basic food commodities were chosen for monitoring: beef, chicken, onion, and chilli, based on the availability of data in terms of tweet mentions and the country-level priorities for food security”. Where there any country-specific priorities that weren’t found on Twitter? It would also be useful to know what limitations there are in using Twitter for this purpose.

On Line 118, the authors write “a tweet is an exact duplication of another (22.9%)”. Exact duplicates are often retweets or people sharing news articles. To what extent does the data come from news articles, and what effect does that have on the approach? If these are indeed news, could one simply have analysed the news articles rather than Twitter?

Line 152, “ beef price jumped to 1.4Mrupiah – 17.5 times larger than official price – for certain days in July2012 from our data” Comment: Is there an error analysis for the spike?

Line 288, “social price is reflected from real-world price and social price may also affect real-world price simultaneously” Comment: To what extent is this feedback loop dependent on the maturation times to grow both commodities?

6. With respect to generality of methods

Line 302, “The nowcast model … predict trends in other essential commodities and across countries”
While the proposed method is attractive, it’s applicability to other countries remains to be seen and so this claim may be too general.

For example, Indonesia, as the authors point out, has very high usage levels of Twitter. For this reason, projects like Petajakarta (using Twitter to gauge flooding levels in Indonesia) have been very successful (https://petajakarta.org/banjir/en/research/; From Social Media to GeoSocial Intelligence: Crowdsourcing Civic Co-management for Flood Response in Jakarta, Indonesia T Holderness, E Turpin. Social Media for Government Services, 115-133).

Similarly, the modelling approach in this paper relies on the heavy usage of Twitter by the community, in this case to report on food prices. This may not be the case in other countries where the predominant online social network is not Twitter (for example, Facebook, which has no search facility). Therefore, while the approach is an attractive one and possibly useful in scenarios outside of Indonesia, one may want to be cautious regarding claims about the generality of the approach.

Minor Corrections:

Line 28, Grammatical or Spelling error:
is largely contributed -> is largely due

Line 30, Grammatical or Spelling error:
Nonetheless, because commodity price and in particular food insecurity in developing regions is extremely dynamic, the ability to track market status quickly at times it is harder to predict is all the more a critical challenge (Gouel, 2013).
->
Nonetheless, because commodity price and, in particular, food insecurity in developing regions is extremely dynamic, the ability to track market status quickly is, at times, hard, making prediction an all the more a critical challenge (Gouel, 2013).

Line 35, Suggested change
Technology enables developing countries to reach a level of financial data access that is available in more developed economies
->
[Comment: Assuming you mean mobile technologies]
Such technology (mobile) enables developing countries to reach a level of financial data access that is available in more developed economies.

Line 38, Grammatical or Spelling error:
price indexes-> price indices

Line 51, Grammatical or Spelling error:
people behavior in crisis situations -> people’s behaviour in crisis situations

Line 70, Grammatical or Spelling error:
Twitter was chosen as data source -> Twitter was chosen as a data source

Line 71, Grammatical or Spelling error:
Indonesia is one of most tweeting countries -> Indonesia has one of the highest adoption rates for Twitter

Line 75, Grammatical or Spelling error:
from the Indonesia region -> from the Indonesian region

Line 76, Suggested change
Another filter on numerical test was also used … -> A (numerical) model parameter was also used …

Line 78, Grammatical or Spelling error:
In the second step, a statistical model was then built to accurately estimate food prices for each commodity in the way OSN data could assist official statistics published by the Indonesian government.
->
In the second step, a statistical model, using OSN data, was then built to accurately estimate food prices for each commodity in order to assist with the publication of official statistics by the Indonesian government.

Line 94, Grammatical or Spelling error:
there is enough online signals -> there are enough online signals

Line 111, Suggested change
“Keyword combination for tweet collection: ( Commodity Names ) AND ( Price Values ) AND ( Price Units | Commodity Units )”
[Comment: This information seems to occur unexpectedly and could do with better introduction.]

Line 120, Grammatical or Spelling error:
Mostly major accounts -> Most major accounts

Line 121, Grammatical or Spelling error:
This finding infers that -> This finding indicates that

Line 136, Grammatical or Spelling error:
the average number of tweets per account to be 2.73 -> the average number of tweets per account is 2.73

Figure 1, Grammatical or Spelling error:
while solid red line is government official price in identical period.
-> while the solid red line is the government official price for the identical period.

Line 142, Grammatical or Spelling error:
price rising period than price decreasing period -> a price-rising period compared to a price-decreasing period.

Line 161, Grammatical or Spelling error:
that of the yesterday’s -> that of yesterday’s

Line 181, Grammatical or Spelling error:
nowcast model assumes there is no price changing. -> nowcast model assumes there is no change in price.

Line 194, Grammatical or Spelling error:
within certain bound. -> within a certain bound.

Line 206, Grammatical or Spelling error:
perform poor when the data is multi-modal, -> performs poorly when the data is multi-modal,
[Comment: defining “multi-modal” may be useful for the reader.]

Line 262, Suggested change:
[Comment: An added figure and an explanatory paragraph highlighting the two data sets being correlated (presumably, price predictions in time for the held-out data) would be useful for the reader.]

Line 271, Suggested change:
According to the concept that suggested by Granger (Granger, 1969) … -> Granger (1969) argues that for two simultaneously measured time series …

Line 281, Suggested change:
We find that nowcasted time series for every target commodity have greater significance level of causal influences.
[Comment: this is unclear, please rephrase.]

Line 292, Grammatical or Spelling error:
help predict economic segments -> helps predict economic behaviour

Line 329, Grammatical or Spelling error:
Another kind of data issues to be considered -> Another data issue to be considered

Line 458,
Vieweg, S., Olteanu, A., and Castillo, C. (2015). What to expect when the unexpected happens: Social 459 media communications across crises. Proceedings of CSCW 2015 (forthcoming).
[Comment: presumably a 2015 is no longer forthcoming]

---

## Round 0.2 · Minor Revisions

This version of the manuscript has been significantly improved. I enjoyed reading the clearer prose and richer details.

I encourage the authors to address further comments from reviewer 2 and below. I expect the manuscript to be ready for publication after successfully addressing theses.

Several suggestions for presentation and reflection:

The proposed model is 'unsupervised', or tuned on separate data, and intuitive -- this is great.

But I wish I know the rationales for the proposed model design: does specific components come from economics literature, or experience, or something else? if the reader were to design a similar model (e.g. iron ore price in Africa, or lentil in India), which parts should she adopt, which part should be changed?

Table 4: suggest shading the table cells to correspond to the value in that cell -- this will make the table a lot easier to read. e.g. can use an annotated heatmap
https://seaborn.pydata.org/examples/heatmap_annotation.html

Figure 5: it's great that the proposed model performs well. However, I (and the readers) would like to see some explanation / observations about why some of the baselines systematically under-predict, such as ARIMA for Onion and KDE for Chilli.

To me the important take-aways are not only in the specific model presented, but also why they work, plus when/why the alternative model break down. Could it be due to data quantity, or the nature of price change, or something else?

·

Basic reporting

I am satisfied that the authors have addressed all the issues relating to "Basic reporting" as detailed in my original review of this paper (the changes to the paper are summarised in the Rebuttal).

Experimental design

I am satisfied that the authors have addressed all the issues relating to "Experimental design" as detailed in my original review of this paper (the changes to the paper are summarised in the Rebuttal).

Validity of the findings

I am satisfied that the authors have addressed all the issues relating to "Validity of the findings" as detailed in my original review of this paper (the changes to the paper are summarised in the Rebuttal).

Additional comments

I believe that the revised paper has addressed all of my main concerns in the previous review, and therefore should be accepted as is. Congratulations to the authors for tackling an important issue using novel methods. I hope to see more of this kind of work in the future.

Reviewer 2 ·

Basic reporting

The authors have addressed the reviewer comments adequately. The introduction and motivation for the work is clearer.

Experimental design

The authors response argues that clarification has been added about (1) how the data was sourced and filtered; (2) how the data was used.

The changes for (1) are satisfactory.

For (2) although this is a largely unsupervised approach (except for the delta parameter), it would still be useful to clarify if the subset of data used to tune parameters (which is 80% of the first 3 months) was separate to the data used to calculate the daily prediction error and the correlation coefficient results in Figure 5 and Table 3.

That is, for these reported results, are the metrics calculated over the 15 months data except for the developmental training data?

The authors do note that all compared models use the same data splits for evaluation (so the comparison between models could be fair). The only concern is whether the report errors and correlation coefficients are higher that what they would be on completely held-out data, which will be the case in a real-world application. (One would expect that this is unlikely given that only the delta parameter is determined using the development data, otherwise the approach is largely unsupervised.)

If this is specified anywhere in the article, it should be clarified again around line 300.

Validity of the findings

As mentioned in #2 above, the only remaining concern to be addressed is whether the reported correlation coefficients are indicative of what one would expect in a real-world application.

Additional comments

The authors have made great progress with the article and it is nearly ready for publication. The only outstanding substantial issue is to clarify which subset of the 15 months worth of data the performance metrics were calculated on. Ideally, the development set was not used in reporting these correlations, otherwise the results may be artificially elevated (at least for the first 3 months of the 15 months of data). Again, one expects the difference not to be large since there would be an addition 12 months of data; still this is a point to be addressed in the article.

One suggestion to tackle this problem would be for the authors to describe the approach as one that requires some initial data set is needed to calculate delta, after which the approach can be used in an unsupervised manner on unseen data.

Minor changes:
Page 1, citation mistake:
line 41
Content: " Premise (Premise, 2016)." -> remove name from parenthesis

Page 2, grammatical or spelling mistake:
line 86
Content: " therefore we could clarifies that the price"
Comment: therefore we can verify that the price

Page 2, Suggested change:
line 97
Content: "This paper also describes the effect of several important social network-wide variables, via testing the robustness of the model under data scarcity conditions and by modeling user-level credibility to suggest an enhanced sampling strategy."
Comment: This paper also describes the effect of several important social network-wide variables, such as (1) testing the robustness of the model under data scarcity conditions, and (2) modeling user-level credibility to suggest an enhanced sampling strategy.

Page 3, grammatical or spelling mistake:
line 116
Content: "monitoring in consultantation with the Ministry of"
Comment: monitoring, defined in consultation
with the Ministry of ...

Page 4, grammatical or spelling mistake:
line 144
Content: "Tweet data contain noisy information and need to be cleaned prior to analysis"
Comment: Tweet data contains noisy information (for example, ...) and needs to be cleaned prior to analysis.

Page 4, Suggested change:
line 144
Content: "We employed the following measures in data cleaning. First involves removing ambiguity in meaning"
Comment: For example, we use filters to remove data that is ambiguous in meaning.

Page 4, Highlight (Yellow):
line 147
Content: "5% of the price quotation data"
Comment: Please clarify, at this point, that this 5% is removed (this is stated later but would be helpful to the reader to state here)).

Page 4, Suggested change:
line 152
Content: " removing redundant messages or spam bots."
Comment: removing redundant messages from spam bots.

Page 4, grammatical or spelling mistake:
near line 163
Content: "Meet Ad"
Comment: Meat Ad

Page 4, Suggested change:
line 165
Content: " and it seems not reasonable to assign more credibility to bot-tweeted information based on its proportion of volume than human-tweeted information"
Comment: (End of sentence) . Correspondingly we do not want to assign more credibility to bot-tweeted information based on its proportion of volume than human-tweeted information.

Page 4, grammatical or spelling mistake:
line 172
Content: " a tweet is an exact duplication of another "
Comment: a tweet is an exact duplicate of another

Page 5, Suggested change:
Content: " data sensitization"
Comment: Define sensitization

Page 12, grammatical or spelling mistake:
near line 399
Content: "Figure 7B "Follwers"
Content: Followers

---

## Round 0.3 · accepted · Accept

I appreciate the authors' efforts in clarifying the manuscript and improving the presentation. I recommend accepting this manuscript for publication.

Regarding the basis of model design, I have one more comment / suggestion.

The term "stationary" seem to be used in ways that confounds what it means, and it is worth being precise:

When the authors say that price time-series are non-stationary -- if we interpreted in the text-book sense of stationary processes (i.e. join distributions does not change, hence any means and co-variances also do not change) -- this is a natural and necessary assumption. Furthermore, does not distinguish the proposed and the comparison methods (e.g. ARIMA) as both are non-stationary.
https://en.wikipedia.org/wiki/Stationary_process

If interpreted in the more-relaxed sense, e.g. time-homogenous Markov chains are considered to have "stationary" parameters, then both the proposed and ARIMA will fall into the same class.

This left me perplexed in what exact intuition the statement in line 215 to 217 tries to explain. lines 316-318 are also confused in similar ways. Defining what "stationary" means in this paper, and then referring to it consistently would help.